# Arteriosclerosis Derived from Cutaneous Inflammation Is Ameliorated by the Deletion of IL-17A and IL-17F

**DOI:** 10.3390/ijms24065434

**Published:** 2023-03-12

**Authors:** Takehisa Nakanishi, Shohei Iida, Junko Maruyama, Hayato Urushima, Masako Ichishi, Yoshiaki Matsushima, Kento Mizutani, Yuichi Nakayama, Kyoko Sugioka, Mai Nishimura, Ai Umaoka, Yoichiro Iwakura, Makoto Kondo, Koji Habe, Daisuke Tsuruta, Osamu Yamamoto, Yasutomo Imai, Keiichi Yamanaka

**Affiliations:** 1Department of Dermatology, Mie University Graduate School of Medicine, 2-174 Edobashi, Tsu 514-8507, Japan; 2Department of Clinical Engineering, Suzuka University of Medical Science, Suzuka 510-0293, Japan; 3Department of Anatomy and Regenerative Biology, Osaka Metropolitan University Graduate School of Medicine, Osaka 545-8585, Japan; 4Department of Oncologic Pathology, Mie University Graduate School of Medicine, 2-174 Edobashi, Tsu 514-8507, Japan; 5Center for Animal Disease Models, Research Institute for Biomedical Sciences, Tokyo University of Science, Chiba 278-8510, Japan; 6Department of Dermatology, Osaka Metropolitan University Graduate School of Medicine, Osaka 545-8585, Japan; 7Division of Dermatology, Department of Medicine of Sensory and Motor Organs, Faculty of Medicine, Tottori University, Yonago 683-8503, Japan; 8Imai Adult and Pediatric Dermatology Clinic, 5-1-1 Ebie, Fukushima, Osaka 553-0001, Japan

**Keywords:** inflammatory skin model mouse, cytokine, arteriosclerosis, endothelial cell, atherosclerosis, IL-17A/F

## Abstract

The skin is one of the major immune organs producing large amounts of proinflammatory and inflammatory cytokines in response to internal or exogenous stimuli, inducing systemic inflammation in various internal organs. In recent years, organ damage associated with inflammatory skin diseases such as psoriasis and atopic dermatitis has received increasing attention, and vascular disorder such as arteriosclerosis is one of the serious complications of chronic inflammatory skin diseases. However, the detailed mechanism of arteriosclerosis in dermatitis and the role of cytokines have not been clarified so far. In the current study, using a spontaneous dermatitis model, we investigated the pathophysiology of arteriosclerosis and the treatment option for inflammatory skin conditions. We employed spontaneous dermatitis model mice overexpressing human caspase-1 in the epidermal keratinocyte (Kcasp1Tg). The thoracic and abdominal aorta was investigated histologically. GeneChip and RT-PCR analysis were performed to measure the changes in mRNA levels in the aorta. To elucidate the direct effect on the artery by major inflammatory cytokines, endothelial cells, vascular smooth muscle cells, and fibroblast cells were co-cultured with several cytokines, and mRNA expression levels were measured. In order to observe the efficacy of IL-17A/F in arteriosclerosis, cross-mating with IL-17A, IL-17F, and IL-17A/F deficient mice was performed. Finally, we also measured snap tension in the abdominal aorta in WT, Kcasp1Tg, and IL17A/F-deficient mice. Kcasp1Tg showed a decrease in the diameter of the abdominal aorta compared to wild-type mice. mRNA levels for six genes including *Apol11b*, *Camp*, *Chil3*, *S100a8*, *S100a9*, and *Spta1* were increased in the abdominal aorta of Kcasp1Tg. Some of the above mRNA levels were also increased in the co-culture with major inflammatory cytokines, IL-17A/F, IL-1β, and TNF-α. Dermatitis improved and mRNA levels were partially ameliorated in Kcasp1Tg with IL-17A/F deletion. Arterial fragility was also evidenced in the inflammatory model, but arterial flexibility was revealed in the IL-17A/F deletion model. Severe dermatitis is closely related to secondary arteriosclerosis caused by the persistent release of inflammatory cytokines. The results also proved that treatment against IL-17A and F may ameliorate arteriosclerosis.

## 1. Introduction

A significant interaction between skin inflammation and organ involvement has become aware. Skin is a composite of the immune system that can respond to exogenous and internal stimuli, triggering systemic inflammation. Vascular disease has been reported as a serious complication in severe intractable skin diseases such as psoriasis, atopic dermatitis (AD), and epidermolysis bullosa (EB) [1,2,3,4].

One of the critical roles of epidermal keratinocytes is to trigger local and systemic inflammation by releasing stored cytokines leading to the activation of the immune system and cytokine cascade. Keratinocyte damage can be the trigger to induce systemic inflammation including the remodeling of vascular tissues, potentially resulting in arteriosclerosis [5]. Excessively produced cytokines from the skin eruption spill over into the systemic circulation and affect remote organs. We have previously proved that the over-production of skin-derived inflammatory cytokines results in organ failures such as cardiovascular and cerebrovascular disorders [6,7], systemic amyloidosis [7,8,9,10], impaired sperm motility [11], osteoporosis [12], and even decreased salivary gland function [13]. Sometimes it leads to inflammation of the adipose tissue resulting in the secretion of adipocytokine, and then activated monocytes and lymphocytes infiltrated into the adipose tissue, which may influence the surrounding tissue including the major artery directly [14]. IL-1 and other released pro-inflammatory cytokines derived from skin or activated leukocytes are likely to contribute to so-called arteriosclerosis. In a human trial, endothelial function measured by flow-mediated dilation (FMD) was improved in patients with psoriasis receiving secukinumab, a fully human monoclonal antibody against IL-17 [15]. However, the detailed mechanism of vascular disorder in inflammatory skin conditions has not been elucidated so far, and the therapeutic efficacy of arteriosclerosis is also unknown. We addressed this problem by using keratin-14-driven caspase-1 transgenic (KCASP1Tg) mice [16]. We investigated the association between inflammatory cytokines from the persistent skin lesion and arteriosclerosis and also revealed the effectiveness of medication for arteriosclerosis.

## 2. Results

### 2.1. Histological Analysis

The histological analysis of the thoracic and abdominal aorta was performed. The arterial stricture was detected in KCASP1Tg mice especially at the abdominal aorta compared to wild-type (WT) mice at four months (Figure 1A). No specific abnormalities were detected with H&E, Congo-Red, and Elastica van Ginson staining on the abdominal aorta. Measurement of the circumference of the lumen of the abdominal artery showed a predominant decrease in KCASP1Tg mice compared to WT littermates (Figure 1B). In contrast, the thoracic portion of the aorta showed no difference in the diameter of KCASP1Tg and WT mice, which was similar to the previous report [7].

### 2.2. Observation of the Abdominal Aorta and Peri-Aortic Adipose Tissue Using Electron Microscopy

On electron microscopy, there are no morphological abnormalities in the endothelial cells, smooth muscle, elastic fibers, or fibroblasts of the abdominal aorta’s outer membrane in both KCASP1Tg and WT mice. The morphology of capillaries is also similar compared to that of WT mice. However, in KCASP1Tg mice, the adipocytes developed abnormal lysosomes, and autophagy-like structures (Figure 2A). There were also fat cells with poorly developed mitochondria in KCASP1Tg mice. More inflammatory cells were infiltrated between fat cells in KCASP1Tg compared to WT mice. In addition, no morphological abnormality is observed in the peripheral nerves. The diameter of collagen fibers in the outer membrane is similar between the two groups (Figure 2B).

### 2.3. GeneChip and Real-Time Polymerase Chain Reaction Analysis for the Abdominal Aorta

The mRNA was extracted from the abdominal aorta of 4-month-old KCASP1Tg and WT mice, and GeneChip analysis was performed. There were several upregulated genes in KCASP1Tg mice including *apolipoprotein L 11b (Apol11b)*, *cathelicidin antimicrobial peptide (Camp)*, *chitinase 3-like 3 (Chil3)*, *S100 calcium-binding protein a8 (calgranulin A, S100a8)*, *S100 calcium-binding protein a9 (calgranulin B, S100a9)*, and *Spectrin alpha, erythrocytic 1 (Spta1)* (Figure 3A). We did not see any significant changes in the mRNA expression of the pro-fibrotic cytokine TGF-β1 and the anti-fibrotic cytokines IFN-γ and TNF-α in the aorta of KCASP1Tg mice or WT. The whole GeneChip gene expression data has been made publicly available in a GEO repository (GSE226446). 

To confirm the upregulation in mRNA expression levels, we performed RT-PCR for *Apol11b*, *Camp*, *Chil3*, *S100a8*, *S100a9*, and *Spta1*. The genes for *Camp*, *Chil3*, *S100a8*, *S100a9*, and *Spta1* were upregulated in abdominal aorta from KCASP1Tg compared to WT mice (Figure 3B), and mRNAs for *Apol11b*, *Camp*, *S100a8*, and *S100a9* were increased in the thoracic aorta (Figure 3B).

### 2.4. Culture of Aorta-Constitute Cells

We measured the changes in mRNA expressions in the cultured endothelial cells, vascular smooth muscle cells, and vascular fibroblasts cultured with inflammatory cytokines. Among the key genes, the mRNAs for *Camp*, *S100a8*, and *S100a9* were significantly increased especially in endothelial cells when co-cultured with a mixture of TNF-α, IL-1β, and IL-17A/F (Figure 4). The culture systems of vascular smooth muscle cells and vascular fibroblasts did not give constant and stable results.

### 2.5. The Effect of Arteriosclerosis by IL-17 Deficiency

To investigate the effect of IL-17A, F, and A/F in arteriosclerosis, the circumference of the lumen of the artery was measured in IL-17A-/KCASP1Tg, IL-17F-/KCASP1Tg, IL-17AF-/KCASP1Tg mice and also IL-17A-, IL-17F-, IL-17AF- mice. The stricture of the lumen of the artery was recovered at the abdominal portion in KCASP1Tg mice when crossing with IL-17KO mice. Mice lacking IL17A/F deficient mice tended to have an enlarged peri-arterial diameter (Figure 5A). The mRNA expression level from the abdominal aorta was partially ameliorated in the IL-17KO-crossed KCASP1Tg mice (Figure 5B).

### 2.6. Snapping Tension in the Abdominal Aorta

KCASP1Tg mice showed marginal fragility compared to WT mice similar to the previous report [7]. A deficiency of IL-17A/F showed rather enhanced vascular firmness and elasticity compared to WT mice (Figure 6).

## 3. Discussion

In the persistent dermatitis model, arteriosclerosis was detected especially at the abdominal aortic portion probably due to the direct effects of adipocytokines produced by the adipose tissue surrounding the abdominal artery (perivascular adipose tissue; *PVAT*) and the large amounts of inflammatory cytokines produced in the inflamed skin coming into the systemic circulation causing endothelial damage. Electron microscopy study revealed poorly developed mitochondria in the adipocytes with autophagy-like structures and lysosomes in KCASP1Tg mice, consistent with the fact that adipocytes are inflamed under the influence of cytokines or by WBCs infiltrating adipose tissue [7,14]. GeneChip and RT-PCR analysis revealed increased key signals during the process of arteriosclerosis in KCASP1Tg. Some of these signals were also enhanced in the co-culture with major inflammatory cytokines, IL-17A/F, IL-1β, and TNF-α. Narrowing of arteries in KCASP1Tg was improved when KCASP1Tg mice were mated with IL-17A, F, and A/F knock-out mice accompanied by decreased key signals in part. IL-17A, F, and A/F deleted mice also had larger arterial perimeters than WT mice, suggesting that excess IL-17A and F may lead to arterial narrowing. The snap tension in the abdominal aorta revealed the arterial wall strength in IL17A/F-deficient mice.

Severe dermatitis is closely related to secondary arteriosclerosis mainly caused by the persistent release of inflammatory cytokines. Although further research is needed, the active control of skin inflammation may be essential to prevent secondary arteriosclerosis. This study defined atherosclerosis in terms of arterial lumen diameter and vulnerability. The pathologies of aortic sclerotic changes were ameliorated in mice with IL-17A/F deletion, revealing that the vascular phenotype can be recovered by inhibiting IL-17A/F. In the abdominal aorta, stenosis was found in the inflammatory model, and its improvement was demonstrated by IL-17 deletion. In the thoracic aorta, on the other hand, the circumference of the artery was unchanged in the inflammatory model, as shown in Figure 1. However, the expression levels of genes predicted to be involved in arteriosclerosis were upregulated in the thoracic aorta of the inflammatory model and improved in the IL-17 deficient model (Figure 2). This finding suggests that subclinical stiffening is progressing due to inflammation. The PVAT is not present around the thoracic aorta; therefore, we speculated that vascular endothelial cells are affected by inflammatory cytokines in the bloodstream.

GeneChip analysis revealed increased six key genes during the process of arteriosclerosis including *Apol11b*, *Camp*, *Chil3*, *S100a8*, *S100a9*, and *Spta1*. Classically the function of apolipoprotein L 11b (*Apol11b*) is thought to be involved in lipid transport and metabolism [17], which is expressed under inflammatory conditions in myeloid and endothelial cells [18]. Apol11b is found in high-density lipoprotein complexes that play a central role in cholesterol transport. The cholesterol content of membranes is important in cellular processes such as modulating gene transcription and signal transduction [19]. Cathelicidin antimicrobial peptide (*Camp*) is a polypeptide that is primarily stored in the lysosomes of macrophages and polymorphonuclear leukocytes. Members of this family react to pathogens by disintegrating, damaging, or puncturing cell membranes. Cathelicidin serves a critical role in mammalian innate immune defense against invasive bacterial infection [20]. Cathelicidin peptides have been isolated from many different species of mammals, and are mostly found in neutrophils, monocytes, mast cells, dendritic cells, epithelial cells, and human keratinocytes [21,22], and the expression increases when injured by external factors, such as trauma, inflammation, or infection. Chitinase 3-like 3 (*Chil3*) is one of the chitinase-like proteins (CLPs), and inactive CLPs are established markers of immune activation and pathology, yet their functions are essentially unknown [23]. CLPs induce the production of pro- and anti-inflammatory cytokines and chemokines and are potential modulators of the inflammatory tumor microenvironment [24]. S100 calcium-binding protein A8 (calgranulin A, *S100a8*) and S100 calcium-binding protein A9 (calgranulin B, *S100a9*) are Ca2+ binding proteins belonging to the S100 family. S100A8/A9 is constitutively expressed in neutrophils and monocytes. During inflammation, S100A8/A9 is released actively and exerts a critical role in modulating the inflammatory response by stimulating leukocyte recruitment and inducing cytokine secretion [25]. Extracellular S100A8/A9 interacts with the pattern recognition receptors Toll-like receptor 4 (TLR4). Finally, spectrin alpha, erythrocytic 1 (*Spta1*) has been identified as one of the causative genes of Hereditary spherocytosis [26], and its role as a mediator transport functions.

In the GeneChip analysis, the above six genes were raised as candidates, among which five genes except *Apol11b* could also be confirmed by RT-PCR. Considering the results of the most sensitive vascular endothelium in culture, *Camp*, *Chil3*, and *S100a8* may be considered the most effective markers of ongoing arteriosclerosis. The leakage of several kinds of inflammatory cytokines from injured keratinocytes induces arteriosclerosis obliterans (ASO) [27]. In many animal models of ASO, hyperlipidemia occurs prior to atheroma formation. Interestingly, KCASP1Tg mice showed no hyperlipidemia nor atheroma plaque formation, suggesting that, in vivo, skin inflammation may induce arteriosclerosis without atheroma formation. The results of the present study provide evidence supporting the association of severe skin inflammation with secondary arteriosclerosis. This is not limited to skin diseases but also to other autoinflammatory disorders.

In the present study, the deficiency of IL-17A, IL-17F, or IL-17A/F improved the vessels’ diameter and their elasticity. Since the increase in vascular diameter was greater than in normal mice, IL-17 may be an essential cytokine to some extent in terms of maintaining homeostasis. However, in daily clinical practice, the main treatment is to decrease IL-17 in the body using neutralizing antibodies, so we may not think that such concerns are necessary. In addition to IL-17 and IL-1, many other types of cytokines are produced in mice and actual dermatitis, and blood vessels are affected by all of them, so further studies focusing on other cytokines are needed.

## 4. Materials and Methods

### 4.1. Mouse Study

Four months-old female transgenic mice in which keratinocytes specifically overexpress the human caspase-1 gene with the K14 promoter, designated as KCASP1Tg, were used in this study [16]. C57BL/6 littermate mice (WT) were used as controls. Animal care was performed according to current ethical guidelines, and the experimental protocol was approved by the Mie University Board Committee for Animal Care and Use (#22-39-5-1).

### 4.2. Tissue Sampling, Histological Analysis, and Observation of the Abdominal Aorta and Adipose Tissue Using Electron Microscopy

All mice were subjected to euthanasia with CO_2_ or pentobarbital. Thoracic and abdominal specimens were fixed in 10% buffered neutral formaldehyde and embedded in paraffin. Histological sections were 6-µm thick and stained with hematoxylin and eosin (H&E), Congo-red, and Elastica van Gieson stain (EVG, *n* = 7, each group). Furthermore, electron microscopy sampled and observed the abdominal aorta after reflux fixation of inflamed mice and normal control mice.

### 4.3. GeneChip Analysis and Real-Time Polymerase Chain Reaction (Real-Time PCR)

GeneChip analysis and RT-PCR were performed to measure the changes in mRNA levels in the thoracic and abdominal aorta. Total RNA was extracted from 16 weeks old mice using Tri Reagent (Molecular Research Center, Cincinnati, OH, USA). The RNA concentration was measured using a NanoDrop Lite spectrophotometer (Thermo Fisher Scientific, Waltham, MA, USA), and one µg total RNA was converted to cDNA using High-Capacity RNA-to-cDNA Kit (Applied Biosystems, Foster City, CA, USA). GeneChip analysis was performed at Takara Bio (Shiga, Japan). RNA profiles in abdominal aorta samples were conducted in eight different mice (*n* = 8). For target synthesis and hybridization, the work was performed using methods recommended by Thermo Fisher Scientific. The labeling protocol was performed by that biotinylated cRNA were synthesized by GeneChip 3′ IVT PLUS Reagent Kit (Affymetrix, Santa Clara, CA, USA) from 250 ng total RNA according to the manufacturer’s instructions. Biotinylated cRNA yield was checked with the NanoDrop ND-2000 Spectro-photometer. Hybridization protocol is that following fragmentation, 10 µg of cRNA were hybridized for 16 h at 45 °C on GeneChip™ Mouse Genome 430 2.0 Array. Arrays were washed and stained in the GeneChip Fluidics Station 450 (Affymetrix). Arrays were scanned using GeneChip Scanner 3000 7G. Single Array Analysis was calculated by Microarray Suite version 5.0 (MAS5.0) with Affymetrix default setting and global scaling as the normalization method. The trimmed mean target intensity of each array was arbitrarily set to 500. The data was imported into the Subio Platform (Subio Inc., Tokyo, Japan). Signals <1 were replaced with 1, log2 transformed, and then mean-subtracted by each probe set to obtain the log ratio against the average of the expression patterns. 

Samples were classified into two groups, WT and KCASP1Tg mice. Probe sets in both groups whose detection values were absent in half of the samples were removed. Finally, unvarying probe sets, whose log ratios were between −1 and +1 in all samples, were filtered out to obtain the final quality-controlled 888 probe sets. Among them, 16 probe sets were further selected.

The TaqMan Universal PCR Master Mix II with UNG (Applied Biosystems, Waltham, MA, USA) was used to measure the mRNA expression of apolipoprotein L 11b (Apol11b, Mm03992571_s1), cathelicidin antimicrobial peptide (Camp, Mm00438285_m1), chitinase-like 3 (Chil3, Mm00657889_mH), S100 calcium-binding protein A8 (S100a8, Mm00496696_g1), S100 calcium-binding protein A9 (S100a9, Mm00656925_m1), spectrin alpha, and erythrocytic 1 (Spta1, Mm01315340_m1). Gryceraldehyde-3-phosphate dehydrogenase (GAPDH, Mm99999915_g1) was used as an internal control. All probes were purchased from Applied Biosystems, and also amplification was performed in a LightCycler 96 System (Roche Diagnostics, Indianapolis, IN, USA). The cycling parameters were as follows: 50 °C for 120 s, 95 °C for 600 s, followed by 40 cycles of amplification at 95 °C for 15 s, and 60 °C for 60 s.

### 4.4. Culture of Aorta-Constitute Cells

To elucidate the direct effect on the artery by major inflammatory cytokines, endothelial cells, vascular smooth muscle cells, and fibroblasts were co-cultured with several cytokines, and mRNA expression levels were measured. Cells were purchased from ATCC (Manassas, VA, USA), and cell culture was at 1 × 10^6^/mL of concentration in RPMI 1640 containing 10% FCS, two mM L-glutamine, 100 U/mL penicillin, and 100 µg/mL streptomycin plated in a 24-well culture plate (Coster, NY, USA) with or without 0.1 µg/mL of TNF-α, IL-1β, or IL-17A/F (BioLegend, San Diego, CA, USA), and then incubated for 24 h (*n* = 5, each group).

### 4.5. IL-17A/F Deficiency in KCASP1Tg Mice

To elucidate the role of IL-17A/F in arteriosclerosis, we mated KCASP1Tg mice with IL-17, IL-17F, and IL-17A/F knockout mice [28]. All mice were sacrificed at the age of 16 weeks.

### 4.6. Snapping Tension of Abdominal Aorta

Mice were anesthetized with sodium pentobarbital (50 mg/kg, i.p.), and the abdominal arteries were isolated, gently cleaned of fat and connective tissue, and cut into rings (1 mm in length) as previously reported [7,29]. Rings were suspended vertically between stainless steel hooks in organ baths (20 mL) with modified Krebs–Henseleit solution (room temperature) containing (in mM): NaCl 115, KCl 4.7, CaCl_2_ 2.5, MgCl_2_ 1.2, NaHCO_3_ 25, KH_2_PO_4_ 1.2, and dextrose 10 in order to record the tension. The changes in isometric tension were measured with a force–displacement transducer (TB-651T; Nihon Kohden, Tokyo, Japan) connected to a carrier amplifier (EF601G; Nihon Kohden) and recorded with a pen recorder (WT-645G; Nihon Kohden). The bathing medium was maintained at 37 °C and bubbled continuously with 95% air and 5% CO_2_. Arterial rings were washed and allowed to equilibrate for 30 min. To measure the tension when the aortic ring snapped, the isometric tension was gradually increased (*n* = 7, each group).

### 4.7. Statistical Analysis

Statistical analysis was performed by using PRISM software version 9 (GraphPad, San Diego, CA, USA). A two-group comparison was analyzed with the Mann-Whitney test, and more than three groups were analyzed with the Ordinary one-way ANOVA followed by Tukey’s multiple comparison test. A *p*-value < 0.05 was considered indicative of statistically significant differences.

## Figures and Tables

**Figure 1 ijms-24-05434-f001:**
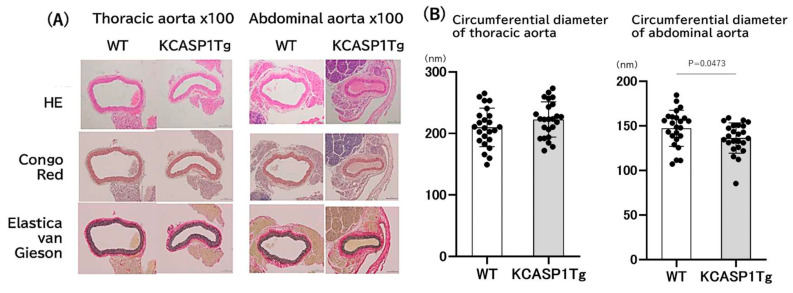
Histological analysis of thoracic and abdominal aorta. The narrowing of the artery was observed in KCASP1Tg mice especially at the abdominal aorta compared to WT mice. Representative pictures of H&E, Congo-Red, and Elastica van Ginson staining were shown (**A**). There were no pathological abnormalities at the thoracic aorta in KCASP1Tg mice. Measurement of the artery’s lumen circumference showed a predominant decrease in the abdominal aorta of KCASP1Tg compared to WT mice. *n* = 24, each group. Two group comparison was analyzed with the Mann-Whitney test (**B**).

**Figure 2 ijms-24-05434-f002:**
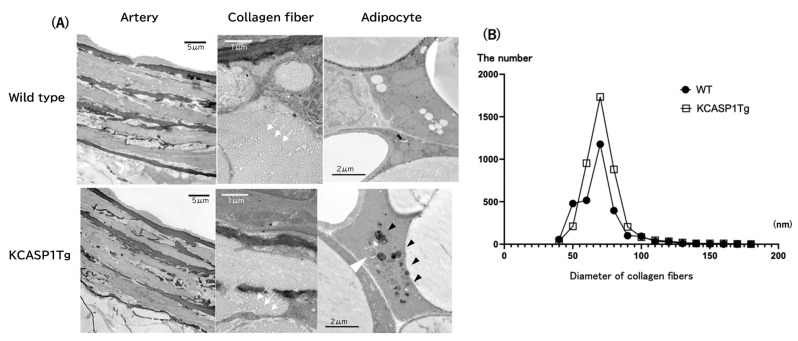
Electron microscopic study for abdominal aorta and peri-aortic adipose tissue. No morphological abnormalities were detected in the endothelial cells, smooth muscle, elastic fibers, or fibroblasts of the outer membrane in both of KCASP1Tg, and WT mice. The white arrow shows each collagen bundle. The adipocytes contained abnormal lysosomes (black arrowhead) and an autophagy-like structure in KCASP1Tg mice (white arrowhead), (**A**). The diameter of collagen fibers in the outer membrane shows a similar pattern between KCASP1Tg, and WT mice (**B**).

**Figure 3 ijms-24-05434-f003:**
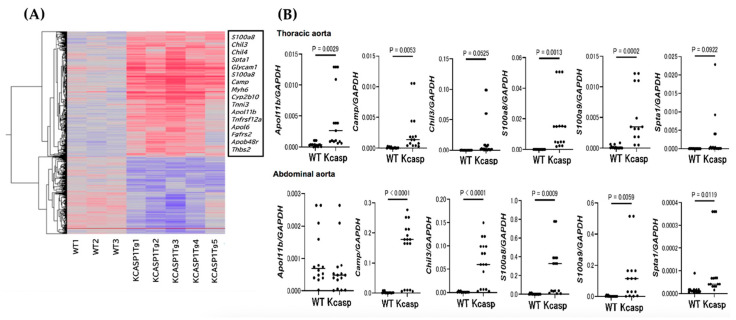
GeneChip and real-time PCR analysis for the thoracic and abdominal aorta. The gene expressions of *Apol11b*, *Camp*, *Chil3*, *S100a8*, and, *S100a9*, *Spta1* have upregulated genes in the abdominal aorta of KCASP1Tg mice (**A**). RT-PCR analysis showed mRNAs for *Camp*, *Chil3*, *S100a8*, *S100a9*, and *Spta1* were upregulated genes in the abdominal aorta of KCASP1Tg mice compared to WT mice. On the other hand, the messages for *Apol11b*, *Camp*, *S100a8*, and *S100a9* were increased in the thoracic aorta. *n* = 15, for each group. Two group comparison was analyzed with the Mann-Whitney test (**B**).

**Figure 4 ijms-24-05434-f004:**
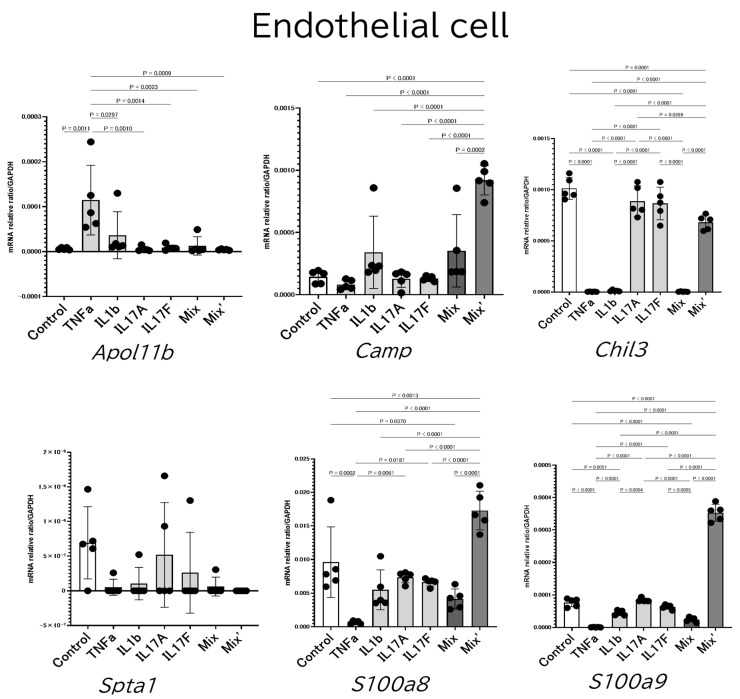
Culture of aorta-constituting cells. We measured the changes in mRNA expressions in the cultured endothelial cells, vascular smooth muscle cells, and fibroblast cells with inflammatory cytokines. The mRNAs for *Camp*, *S100a8*, and *S100a9* were significantly increased especially in the endothelial cell when co-cultured with the mixture of TNF-α, IL-1β, or IL-17A/F. The culture systems of vascular smooth muscle cells and vascular fibroblasts did not provide constant and stable results. Mix = TNF-α, IL-1β, IL-17A. Mix’ = TNF-α, IL-1β, IL-17A, IL-17F. *n* = 5, each group. More than three groups were analyzed with the Ordinary one-way ANOVA followed by Tukey’s multiple comparison test.

**Figure 5 ijms-24-05434-f005:**
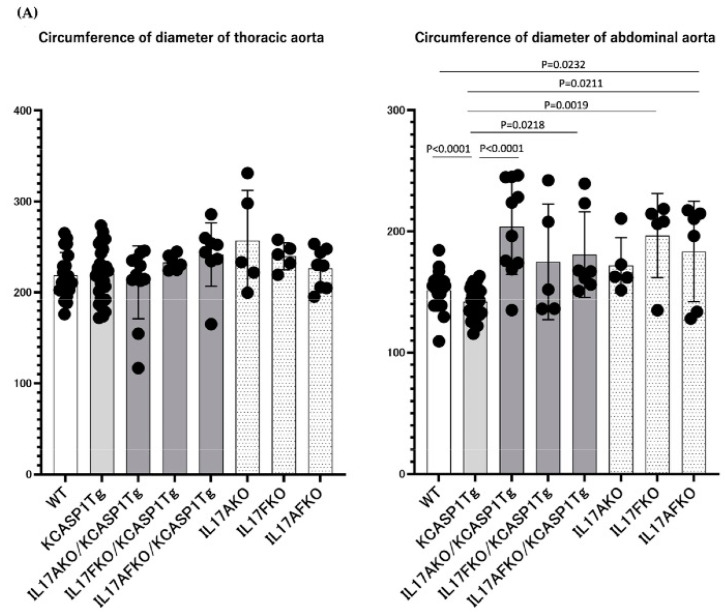
The effect of arteriosclerosis by IL-17 deficiency. The circumference of the lumen of the artery was recovered at the abdominal portion in KCASP1Tg mice when crossing with IL-17A-, IL-17F-, and IL-17AF- mice. In addition, IL-17A-, IL-17F-, and IL-17AF- mice tended to have an enlarged aorta (**A**). The mRNA expression levels for the enhanced gene in KCASP1Tg mice were partially ameliorated in IL-17KO KCASP1Tg mice. *n* = 6, each group. Statistical analysis was performed with the Ordinary one-way ANOVA followed by Tukey’s multiple comparison tests (**B**).

**Figure 6 ijms-24-05434-f006:**
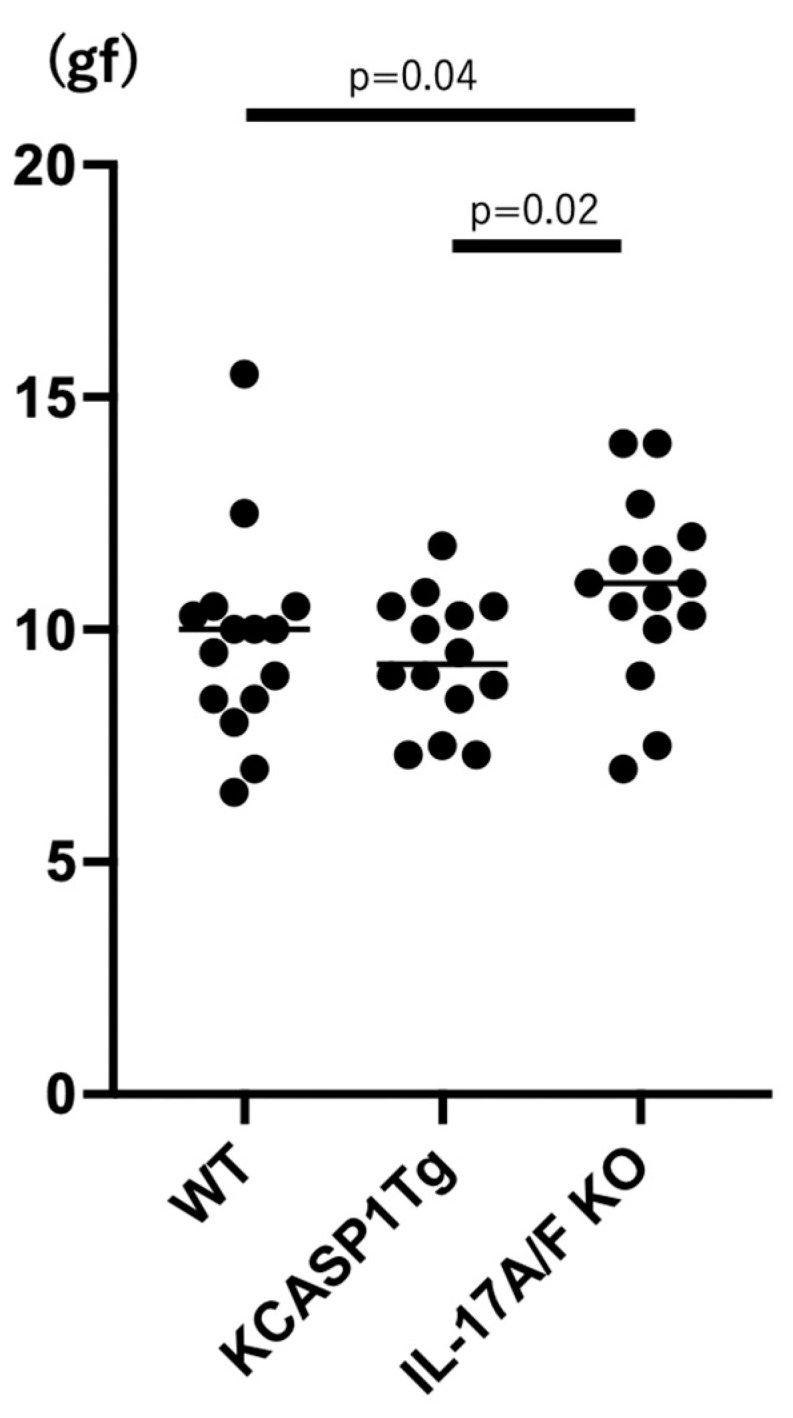
Snapping tension in the abdominal aorta. KCASP1Tg mice showed fragility compared to WT mice. A deficiency of IL-17A/F showed rather enhanced vascular firmness and elasticity compared to WT littermates. N = 15, for each group. The statistical analysis was performed with the Ordinary one-way ANOVA followed by Tukey’s multiple comparison test.

## Data Availability

The whole GeneChip gene expression data has been made publicly available in a GEO repository (GSE226446).

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
