# Peer review of "Arteriosclerosis Derived from Cutaneous Inflammation Is Ameliorated by the Deletion of IL-17A and IL-17F"

_ijms, 2023, doi:10.3390/ijms24065434_

Round 1
Reviewer 1 Report
This study reports the effect of skin inflammation in aorta using a mouse model that spontanoeus dermatitis that expresses caspase-1 in keratinocytes. In general, the mechanisms involved in the development of comorbidities associated to dermatitis are not very well known. So, this study is important and timely. However, the manuscript requires extensive polishing and include much more information to clarify the experimental procedures and design. In general all figures need to be extensively polished and the legends completely rewritten to indicate sample size and statistical analysis. In its present form, it is not possible to know how most of the experiments were performed.
Major points:
1. Section 2.2 and Fig. 2: please, labelled all the structures taht you describe in Results, such as collagen fibers, fat and inflammatory cells, etc. Inflammatory cell infiltration needs quantitation. Autophagy, which is mention in Discussion but not Results, needs to be quantitated using appropriate assays. If not, this has to be removed.
2. Fig. 3: Indicate sample size and statistical analysis rather than repeat the description of the data; already shown in Results. This apply to all figures. Increase size font: it is not possible to read the genes and label to axes. Which tissue did you use for the GeneChip: abdomeinal or tharaci aorta? This is critical, since you found obvious differences between them.
3. Fig. 4:Indicate sample size and statistical analysis in the legend. Increase size font. Endothelial is misspelled. Importantly, there are 2 different mix. There is not information about them in the figure or M&M section. How did you analyze gene expression in the 3 cell population after coculturing them? Fianlly, more biological replicates are required for fibroblasts. There are some samples with 2 replicates.
4. Fig. 5: Indicate sample size and statistical analysis in the legend. Increase size font. You find a lower rescue with double KO compared with single KO for IL17. This does not make sense and need to be appropriately explained.
5. Fig. 6: Indicate sample size and statistical analysis in the legend. Why only the DO KO was used to quantitate the snapping tension of the abdominal aorta, especially when you bserved a better rescue with single KO?
6. Discussion: you mentioned that the identified genes may be good prognosis biomarkers for arteriosclerosis-associated to dermatitis. This would require longitudinal studies in your mouse model. Why did you use 4 month-old mice? I strongly suggest to analysis the expression of these genes at differnt time points together with snapping tension and histological anaysis of abdominal aorta. This will greatly increase the impact of this study.
Minor points:
1. Page, 1st line: This sentence does not make sense: "arterial flexibility was revealed in IL-17A/F deletion"
2. Section 2.1, 1st line: "The arterial stricture was detected"? I suppose you mean the arterial structure was analyzed/studied.
Author Response
Responses to the comments of Reviewer #1
Comments to the Author:
This study reports the effect of skin inflammation in aorta using a mouse model that spontanoeus dermatitis that expresses caspase-1 in keratinocytes. In general, the mechanisms involved in the development of comorbidities associated to dermatitis are not very well known. So, this study is important and timely. However, the manuscript requires extensive polishing and include much more information to clarify the experimental procedures and design. In general all figures need to be extensively polished and the legends completely rewritten to indicate sample size and statistical analysis. In its present form, it is not possible to know how most of the experiments were performed.
Response: Thank you for your critical suggestions. We have reconsidered and corrected each of the comments you pointed out.
Major points:
- Section 2.2 and Fig. 2: please, labelled all the structures taht you describe in Results, such as collagen fibers, fat and inflammatory cells, etc. Inflammatory cell infiltration needs quantitation. Autophagy, which is mention in Discussion but not Results, needs to be quantitated using appropriate assays. If not, this has to be removed.
Response: Thank you for your suggestion. We have indicated collagen fibers and adipocyte in the figure. We have not included the quantification of infiltrating inflammatory cells in the current article, since the previous paper already reported the number using an appropriate assay as shown in the references 7 and 14.
- 3: Indicate sample size and statistical analysis rather than repeat the description of the data; already shown in Results. This apply to all figures. Increase size font: it is not possible to read the genes and label to axes. Which tissue did you use for the GeneChip: abdominal or tharaci aorta? This is critical, since you found obvious differences between them.
Response: Thank you for your suggestion. I have supplemented the sample size and the results of the statistical analysis. I also adjusted the font of the gene and axis labels to make them easier to read. The abdominal aorta was used for GeneChip. Thank you for your critical remarks.
- 4:Indicate sample size and statistical analysis in the legend. Increase size font. Endothelial is misspelled. Importantly, there are 2 different mix. There is not information about them in the figure or M&M section. How did you analyze gene expression in the 3 cell population after coculturing them? Fianlly, more biological replicates are required for fibroblasts. There are some samples with 2 replicates.
Response: Thank you for your suggestion. We have added the sample size and statistical analysis results to the legend. We also corrected a misspelling of endothelium, font size, etc. We added an explanation of mix and mix'. Added more biological replicates for fibroblast and revised the figure. Each of the three cell populations was co-cultured with cytokines (TNFα, IL1b, IL17A, IL17F) alone or mixed (Mix=TNFα, IL1b, IL17A Mix'=TNFα, IL1b, IL17A, IL17F), cells were collected after 24 hours, RNA was collected and converted to cDNA and analyzed for gene expression using realtime-PCR.
- 5: Indicate sample size and statistical analysis in the legend. Increase size font. You find a lower rescue with double KO compared with single KO for IL17. This does not make sense and need to be appropriately explained.
Response: Thank you for the suggestion. We have supplemented sample size and statistical analysis results to legend, adjusted font size. We increased the number of samples as much as possible and re-analyzed the results. As a result, not only single KO, but also double KO showed low expression levels of some genes. At this point, it is suggested that IL17A or F may have some effects at the gene level in arteries by acting alone or in interaction, but we consider that further investigation is necessary.
- 6: Indicate sample size and statistical analysis in the legend. Why only the DO KO was used to quantitate the snapping tension of the abdominal aorta, especially when you bserved a better rescue with single KO?
Response: Thank you for your suggestion. We have added the sample size and statistical analysis in the legend and adjusted the font size. We examined the elasticity of the arterial wall in the abdominal aorta where pathological changes of atherosclerosis were occurring. As noted by reviewer 1, whether IL-17A or IL-17F contributes to the elasticity of the arterial wall is not known. This is an experiment that should be investigated, and we planned to do so. However, it was not possible to prepare sufficient quantities of both IL-17A and IL-17F knockouts for the experiment due to reproductive problems. We plan to make an additional report as soon as the numbers are available. Thank you for your important remarks.
- Discussion: you mentioned that the identified genes may be good prognosis biomarkers for arteriosclerosis-associated to dermatitis. This would require longitudinal studies in your mouse model. Why did you use 4 month-old mice? I strongly suggest to analysis the expression of these genes at differnt time points together with snapping tension and histological anaysis of abdominal aorta. This will greatly increase the impact of this study.
Response: In this inflammation model, skin rash appears at the age of 2 months, peaks at the age of 4 months, and turns into chronic eczema at the age of 6 months. Based on the results of previous papers, including Ref. 7, the clinical symptoms are already fixed at 6 mo. Based on genechip analysis, the current study was initiated. mRNA changes and protein changes should be implemented at the peak of inflammation, and inflammation has disappeared at 6 mo. Since this study was to search for factors contributing to atherosclerosis, we first chose a 4-month-old mouse model of dermatitis. As you indicated, we should consider gene expression by week age, abdominal snap tension, and histopathology as future research topics. Thank you very much for your useful suggestions.
Minor points:
- Page, 1st line: This sentence does not make sense: "arterial flexibility was revealed in IL-17A/F deletion"
Response: Thank you for the suggestion. We have corrected the sentence.
.
- Section 2.1, 1st line: "The arterial stricture was detected"? I suppose you mean the arterial structure was analyzed/studied.
Response: Thank you for the suggestion. Since this is the results section, it is safe to say that arterial stenosis was detected. However, we have added the text for the introduction.
Reviewer 2 Report
This is a clearly written and well-organized manuscript which might be of interest for the audience to read. However, the sample size needs to be increased. Some figures, such as figure 4 and 5, show many groups with n=1 or n=2. This needs to be addressed before publication. In addition to this major concern, some minor typo mistakes need to be corrected such as “endotherial“ in figure 4.
Author Response
Responses to the comments of Reviewer #2
Comments to the Author:
This is a clearly written and well-organized manuscript which might be of interest for the audience to read. However, the sample size needs to be increased. Some figures, such as figure 4 and 5, show many groups with n=1 or n=2. This needs to be addressed before publication. In addition to this major concern, some minor typo mistakes need to be corrected such as “endotherial“ in figure 4.
Response: Thank you for your suggestion. We have re-analyzed and modified the figure by increasing the number of samples as much as possible. Fonts have been adjusted. Also, we have corrected the spelling error of "endotherial" to "endothelial" in Figure 4.
Reviewer 3 Report
The paper by Nakanishi et al., 2023 Arteriosclerosis derived from cutaneous inflammation is ame-liorated by the deletion of IL-17A and IL-17F is discribing spontaneous dermatitis in a mice model wher they overexpressed human caspase-1 in the epidermal keratinocyte (Kcasp1Tg). At the same time they tried to evalute the efficacy of IL-17A/F in arterioscle-rosis, cross-mating with IL-17A, IL-17F, and IL-17A/F deficient mice.
In the text there a few mistakes in writing, gaps missing, dots added, exampel;
Figure 2. electron microscopic study for abdominal aorta and peri-aortic adipose tissueNo mor-
Figure 3. GeneChip and real-time PCR analysis for the thoracic and abdominal aortaThe gene ex-
Endotherial cells (figure 4)
In the figure 5 the values on y-axis are very small.
All this mistakes looks like the paper is very clumsy.
Please check the text and figures!!!
This model of knock in Kcasp1Tg and knock out IL-17A/F and then analysing histologoy and mRNA could have some misleading results. Could be the same gene and histological assay anaysed also with some trigger of inflamation and analysed in comparison to IL-17A/F knockout and Kcasp1Tg?
Author Response
Responses to the comments of Reviewer #3
Comments to the Author:
The paper by Nakanishi et al., 2023 Arteriosclerosis derived from cutaneous inflammation is ame-liorated by the deletion of IL-17A and IL-17F is discribing spontaneous dermatitis in a mice model wher they overexpressed human caspase-1 in the epidermal keratinocyte (Kcasp1Tg). At the same time they tried to evalute the efficacy of IL-17A/F in arterioscle-rosis, cross-mating with IL-17A, IL-17F, and IL-17A/F deficient mice.
In the text there a few mistakes in writing, gaps missing, dots added, exampel;
Figure 2. electron microscopic study for abdominal aorta and peri-aortic adipose tissueNomor-
Figure 3. GeneChip and real-time PCR analysis for the thoracic and abdominal aortaThe gene ex-
Endotherial cells (figure 4)
In the figure 5 the values on y-axis are very small.
All this mistakes looks like the paper is very clumsy.
Please check the text and figures!!!
Response: Thank you for your suggestion. We have reconsidered the text and figures and corrected the mistakes you pointed out. We have also adjusted the fonts and other details of the figures.
This model of knock in Kcasp1Tg and knock out IL-17A/F and then analysing histologoy and mRNA could have some misleading results. Could be the same gene and histological assay anaysed also with some trigger of inflamation and analysed in comparison to IL-17A/F knockout and Kcasp1Tg?
Response: Kcasp1Tg is a mouse model of spontaneous skin inflammation that overexpresses caspase-1 in the epidermal keratinocyte. Inflamed skin overexpresses not only IL-1β and IL-18, which are converted to their active forms by caspase-1, but also many inflammatory cytokines including IL-17. As a result, various visceral complications occur, including atherosclerosis. The current study is an experiment to investigate whether IL-17A/F contributes to the complications. As the reviewer 3 pointed, in Figure 5, the same genes were examined in WT, KCASP1Tg, and IL-17A-, IL-17F-, IL-17AF-KO KCASP1Tg mice. Thank you very much for your important suggestions for our future research.
Round 2
Reviewer 1 Report
Although most concerns were satisfactorily addressed, the problem related to the description of autophagy-like structures in Figure 2 was not addressed. Please, describe this result in section 2.2 rather than the legend to Fig.2 and clearly label them in the figure.
Author Response
Responses to the comments of Reviewer #1
Comments to the Author:
Although most concerns were satisfactorily addressed, the problem related to the description of autophagy-like structures in Figure 2 was not addressed. Please, describe this result in section 2.2 rather than the legend to Fig.2 and clearly label them in the figure.
Response: Thank you for your suggestions. We have labeled autophagy-like structures in Figure 2. We also described it in section 2.2. We appreciate your comment.
Reviewer 2 Report
The authors adressed my main concern by increasing the sample size. I would accept the manuscript in the present form.
Author Response
Responses to the comments of Reviewer #2
Comments to the Author:
The authors adressed my main concern by increasing the sample size. I would accept the manuscript in the present form.
Response: Thank you for your suggestions. By increasing the sample size, trends in events could be presented more clearly in the paper. Thank you for the remarks.